# Mapping Research on Microbial Fuel Cells in Wastewater Treatment: A Co-Citation Analysis

**Tianming Chen** [1,2]**, Chao Zou** [1]**, Jingjing Pan** [1]**, Mansi Wang** [1]**, Liang Qiao** [1,2]**, Feihong Wang** [1,2]**, Qi Zhao** [1]**, Haoyi Cheng** [3]**, Cheng Ding** [1,2] **and Ye Yuan** [1,2,3,*]

1   School of Environmental Science and Engineering, Yancheng Institute of Technology, Yancheng 224051, China; ycchentm@ycit.edu.cn (T.C.); zouchaofighting@163.com (C.Z.); jingjingpan2021@163.com (J.P.); wangmansi1015@126.com (M.W.); SagittariusBK@163.com (L.Q.); wfh82090446@sohu.com (F.W.); zq_18860823620@163.com (Q.Z.); ycdingc@163.com (C.D.)
2   Jiangsu Province Engineering Research Center of Intelligent Environmental Protection Equipment, Yancheng Institute of Technology, Yancheng 224051, China
3   Research Center for Eco-Environmental Sciences, Key Laboratory of Environmental Biotechnology, Chinese Academy of Sciences, Beijing 100085, China; hycheng@rcees.ac.cn
*   Correspondence: yuanye_19840915@163.com

**Abstract:** Microbial fuel cells (MFCs) are promising technologies, aiming at treating different types of industrial and domestic wastewater. In recent years, more and more publications focusing on wastewater treatment have been published. Based on the retrieval of publications from Web of Science Core Collection database, the new emerging trends of microbial fuel cells in wastewater treatment was evaluated with a scientometric analysis method from 1995 to 2020. All publications downloaded from (WOS) were screened by inclusion criteria, and 2233 publications were obtained for further analysis. Document co-citation and burst detection of MFCs in wastewater treatment were analyzed and visualized by software of CiteSpace. Our study indicated that "Environmental Science" is the most popular discipline, while the journal of *Bioresource Technology* published the greatest quantity of articles in the field of MFCs applied wastewater treatment. China and the Chinese Academy of Science are the most productive country and institution, respectively. "Azo dye" has become the new research topic, which indicates the application area and the development of MFCs. The performance of MFCs for wastewater treatment has been widely discussed. The findings of this study may ameliorate the researcher in seizing the frontier of MFCs in wastewater treatment.

**Keywords:** microbial fuel cells (MFCs); wastewater treatment; CiteSpace; quantitative; visualization analysis

## 1. Introduction

Nowadays, the two major global problems facing human beings are energy shortage and environmental pollution [1]. Therefore, great efforts have long been exerted in a simultaneous response to both the energy consumption and water contamination [2]. The wastewater containing pollutants must be treated before being discharged into the environment [3,4]. At present, wastewater treatment is commonly treated with a conventional aerobic activated sludge reactor, anaerobic digester, membrane filtration, ion exchange, adsorption, coagulation, electrolytic reduction and so on [5]. Nevertheless, the high expenditure of energy and the running cost are the two major restraining factors for the current wastewater treatment technologies [6]. In addition, the presence of a large amount of residual generation can lead to secondary pollution among these technologies, which can be deleterious for the environment and ineffective in catching the energy potential from wastewater [7]. Therefore, it is essential to establish a wastewater treatment technology, which is required to be reliable, sustainable, and cost-effective [8].

In recent years, microbial fuel cells (MFCs) have been demonstrated as a promising yet challenging technology that serves a dual purpose: pollutant removal and energy recovery [4,9]. In MFCs, the oxidation of organic compounds occur in an anode chamber catalyzed by microbes, while the electrons are transferred to a cathode in which reduction takes place, generating bioenergy and accomplishing water treatment simultaneously [10–12]. In 2001, it was first proposed to establish the mode of using MFCs to achieve power generation and wastewater treatment, in which starch industrial wastewater was used as the substrate electricity generation [13]. Since then, MFCs have been widely used to treat different types of industrial and domestic wastewater, such as that of molasses [14], yogurt [15], oil refineries [16], beer breweries [17], the chocolate industry [18], the paper industry [19] and so on.

Compared with other wastewater treatment technologies, MFCs have the following significant advantages: (1) direct conversion of substrates energy into electricity, (2) low activated sludge generation, (3) being robust and insensitive to environmental factors (e.g., temperature), (4) absence of gas treatment, (5) without any energy input for aeration, and (6) a widespread application in places lacking electrical infrastructures [20,21]. MFCs have proven to have great potential for industrial applications in several types of wastewater treatment [22]. To date, the number of papers on MFCs in wastewater treatment is increasing. Carlos et al. [6] demonstrated that MFCs can generate electricity and achieve wastewater treatment effectively. However, few works in the literature have focused on the development trend of MFC in wastewater treatment based on bibliometrics and visualization technology. Bibliometric is an effective and useful tool that can evaluate the scientific production and research trends in a specific research field [23]. CiteSpace is an information visualization software developed by Chen Chaomei from Drexel University, which is primarily based on Java language for analysis and visualization of co-citation networks [24,25]. In recent years, the majority of scholars gradually increased attention on it. Li et al. [26] estimated the research front of bioelectrochemical systems by CiteSpace software. Through quantitative and visual analysis of publication information collected from Web of Science, it can provide novel sight on the changing and development trend of MFCs research in wastewater treatment, which is of great significance for researchers to further understand the direction of MFCs for future study and application.

With the aim to explore the key literatures and major research area on MFCs in wastewater treatment, we summarized the published information in this field from the Web of Science Core Collection in the past 25 years (1995–2020) and conducted co-citation analysis to identify the core literatures and investigate the major research area on MFCs in wastewater treatment by CiteSpace. This study estimates the most important and remarkable research trends and hot spots of MFCs usage in wastewater treatment. In addition, this study provides an overview of the development of MFCs usage in wastewater treatment and also offers valuable suggestions to energy and environmental management.

## 2. Materials and Methods

The purpose of this study is to make a visualization and analysis of MFCs usage in wastewater treatment. The literature data were downloaded from Web of Science, using data collected on 16 May 2021. The search formula was TS (Topic Search) = Topic: "Microbial fuel cell" and Topic: "Wastewater treatment". The search period was set as "All years (from 1995 to 2020)". By combining wastewater treatment with MFCs search results, a total of 2233 publications were obtained, and then the full record and cited references were exported to CiteSpace for further analysis. The distribution of annual published documents is presented in Figure 1.

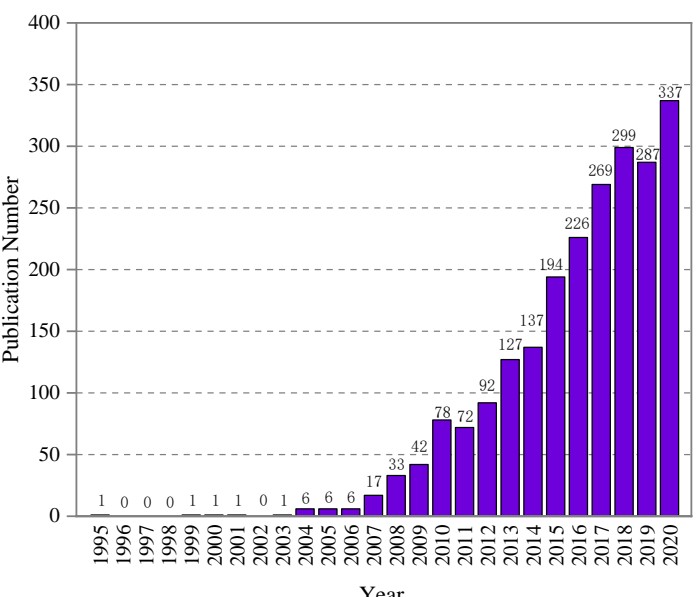

**Figure 1.** Distribution of publications per year on MFCs in wastewater treatment from 1995 to 2020.

The data based on the document set we retrieved were further analyzed by bibliometrics. The published subject categories, journals, co-occurrence analysis (e.g., countries/territories and institutions) document co-citation and burst detection of MFCs in wastewater treatment were conducted and visualized by the software of CiteSpace.

## 3. Current Status of Microbial Fuel Cells (MFCs) in Wastewater Treatment

Based on the document set we retrieved, we concluded that the article related MFCs in wastewater treatment was first reported in 1995. In the last 25 years, the quantity of published documents on MFCs in wastewater treatment increased from 1 in 1995 to 337 in 2020. The number of publications has steadily increased every year. This fully explains the research on MFCs in wastewater treatment getting more and more attention. The distribution of annual citation is shown in Figure 2. There is an obvious upward trend for the annual citation on MFCs in wastewater treatment.

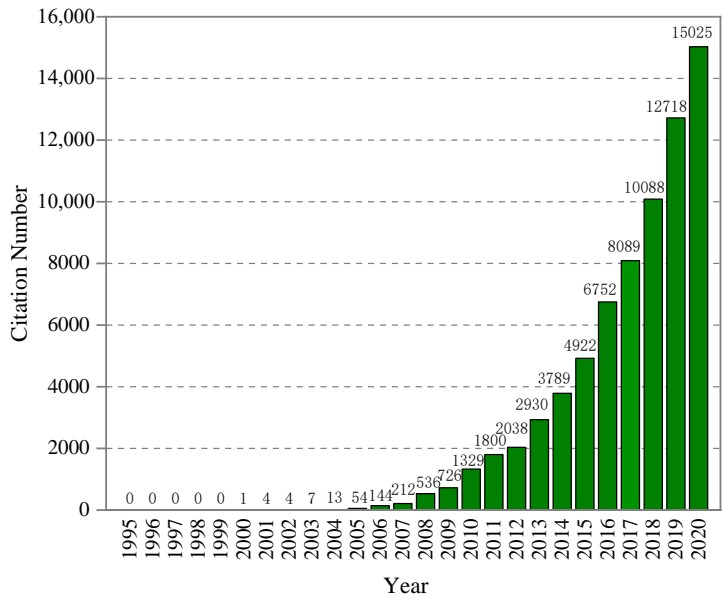

**Figure 2.** Distribution of citations per year on MFCs in wastewater treatment from 1995 to 2020.

As is shown in Figure 3, a total of 2233 publications related to MFCs in wastewater treatment were categorized into 8 document types (Article, Review, Proceeding paper, Early access, Book chapter, Meeting abstract, Editorial material, and Correction). In addition, Articles are the dominant document publication type (comprising 81.0% of the overall documents), followed by Reviews (13.5%), Proceeding papers (7.6%) and other types of publications. Among these 2233 publications related to studies of MFCs in wastewater treatment, they were categorized into more than 50 subjects. The top 10 subject categories on MFCs in wastewater treatment are summarized and listed in Table 1.

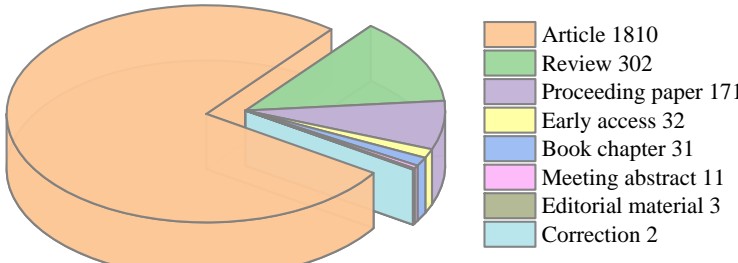

**Figure 3.** Document types on MFCs in wastewater treatment from 1995 to 2020.

**Table 1.** The top 10 subject categories on MFCs in wastewater treatment.

| Subject Categories | Publication Number | The Percentage of Total |
|---|---|---|
| Environmental Sciences | 652 | 29.198 |
| Energy Fuels | 639 | 28.616 |
| Biotechnology Applied Microbiology | 560 | 25.078 |
| Engineering Environmental | 513 | 22.974 |
| Engineering Chemical | 435 | 19.481 |
| Electrochemistry | 277 | 12.405 |
| Agricultural Engineering | 262 | 11.733 |
| Water Resources | 254 | 11.375 |
| Chemistry Physical | 180 | 8.061 |
| Chemistry Multidisciplinary | 168 | 7.524 |

"Environmental Sciences" and "Energy Fuels" were identified as the two most popular subject categories with 652 publications and 639 publications, which contributed 29.198% and 28.616% of the total publications, respectively. "Biotechnology Applied Microbiology" contributed 25.078% of the total publications and occupied the third most popular subject category position, followed by "Engineering Environmental", "Engineering Chemical", "Electrochemistry", "Agricultural Engineering", "Water Resources", "Chemistry Physical", and "Chemistry Multi-disciplinary".

Hundreds of journals have published research related to MFCs in wastewater treatment. Table 2 shows the top 10 journals which produced the largest quantity of publications in the field of MFCs in wastewater treatment. We found that the journal *Bioresource Technology* with 250 publications about MFCs in wastewater treatment published the greatest number of articles and accounts for 11.196% of the total publications. The second productive journal is *International Journal of Hydrogen Energy* with 77 articles followed by *Chemical Engineering Journal*, *Water Research*, *Journal of Power Sources*, *Water Science and Technology*, *Environmental Science & Technology*, *Science of the Total Environment*, *Journal of Chemical Technology and Biotechnology* and *Environmental Technology*.

**Table 2.** The top 10 journals on MFCs in wastewater treatment.

| Journals | Publication Number | The Percentage of Total |
|---|---|---|
| *Bioresource Technology* | 250 | 11.196 |
| *International Journal of Hydrogen Energy* | 77 | 3.448 |
| *Chemical Engineering Journal* | 71 | 3.180 |
| *Water Research* | 70 | 3.135 |
| *Journal of Power Sources* | 54 | 2.418 |
| *Water Science and Technology* | 52 | 2.329 |
| *Environmental Science & Technology* | 45 | 2.015 |
| *Science of the Total Environment* | 45 | 2.015 |
| *Journal of Chemical Technology and Biotechnology* | 39 | 1.747 |
| *Environmental Technology* | 37 | 1.657 |

The research on MFCs in wastewater treatment in different countries/territories attracted different degrees of attention. The top 10 most prolific countries on MFCs in wastewater treatment are listed in Table 3. From Table 3, China was identified to publish the largest number of publications on MFCs in wastewater treatment, followed by the U.S.A., India, South Korea, Spain, Australia, Malaysia, England, Iran and Italy.

**Table 3.** The top 10 most productive countries on MFCs in wastewater treatment.

| Country/Territories | Number | The Percentage of Total |
|---|---|---|
| Peoples R China | 756 | 33.856 |
| U.S.A. | 392 | 17.555 |
| India | 333 | 14.913 |
| South Korea | 135 | 6.046 |
| Spain | 111 | 4.971 |
| Australia | 94 | 4.210 |
| Malaysia | 83 | 3.717 |
| England | 81 | 3.627 |
| Iran | 60 | 2.687 |
| Italy | 60 | 2.687 |

As is known to all, China and the U.S.A. are currently the world's top two economies. In order to cope with the increase in global energy demand and the depletion of fossil fuels, scientists and researchers in the two countries have paid much attention to eco-friendly and economically feasible renewable energy sources. As a promising environmentally friendly technology, MFCs can directly convert organic energy into electrical energy, thereby achieving simultaneous power generation and waste recycling. Therefore, countries, led by China and the U.S.A., have conducted a lot of research on MFCs in wastewater treatment related research.

Table 4 indicates that Chinese Academy of Sciences from China is the most productive institution on MFCs in wastewater treatment (98 publications). Harbin Institute of Technology from China is the second most productive institutions with 92 publications followed by Pennsylvania State University (U.S.A.), Virginia Polytechnic Institute and State University (U.S.A.), Indian Institute of Technology (India), Tsinghua University (China), Dalian University of Technology (China), Nankai University (China), Indian Institute of Chemical Technology (India), and University of Science and Technology of China (China). Among the above 10 mentioned institutions, 6 belonged to China, with 2 from U.S.A. and anther 2 from India. Though South Korea, Spain, Australia, Malaysia, England, Iran and Italy were the most productive countries on MFCs in wastewater treatment, there are no institutions from these countries involved in the MFCs-related research.

**Table 4.** The top 10 most productive institutions on MFCs in wastewater treatment.

| Institutions (Country) | Number | The Percentage of Total |
|---|---|---|
| Chinese Academy of Sciences (China) | 98 | 4.389 |
| Harbin Institute of Technology (China) | 92 | 4.120 |
| Pennsylvania State University (U.S.A.) | 65 | 2.911 |
| Virginia Polytechnic Institute and State University (U.S.A.) | 65 | 2.911 |
| Indian Institute of Technology (India) | 55 | 2.463 |
| Tsinghua University (China) | 46 | 2.060 |
| Dalian University of Technology (China) | 42 | 1.881 |
| Nankai University (China) | 35 | 1.567 |
| Indian Institute of Chemical Technology (India) | 31 | 1.388 |
| University of Science and Technology of China (China) | 31 | 1.388 |
| Chinese Academy of Sciences (China) | 98 | 4.389 |

## 4. Document Co-Citation Analysis on MFCs in Wastewater Treatment

Document co-citation analysis and the citation frequency were simultaneously conducted to investigate the important articles which have had a significant impact on MFCs in wastewater treatment, and the main research area was also estimated.

### 4.1. Research Clusters on MFCs in Wastewater Treatment

The co-citation network and the main clusters on MFCs in wastewater treatment were investigated and visualized by the software of CiteSpace (Figures 4 and 5). There are 18 co-citation clusters labeled with citers in the overall network and shown in Figure 5.

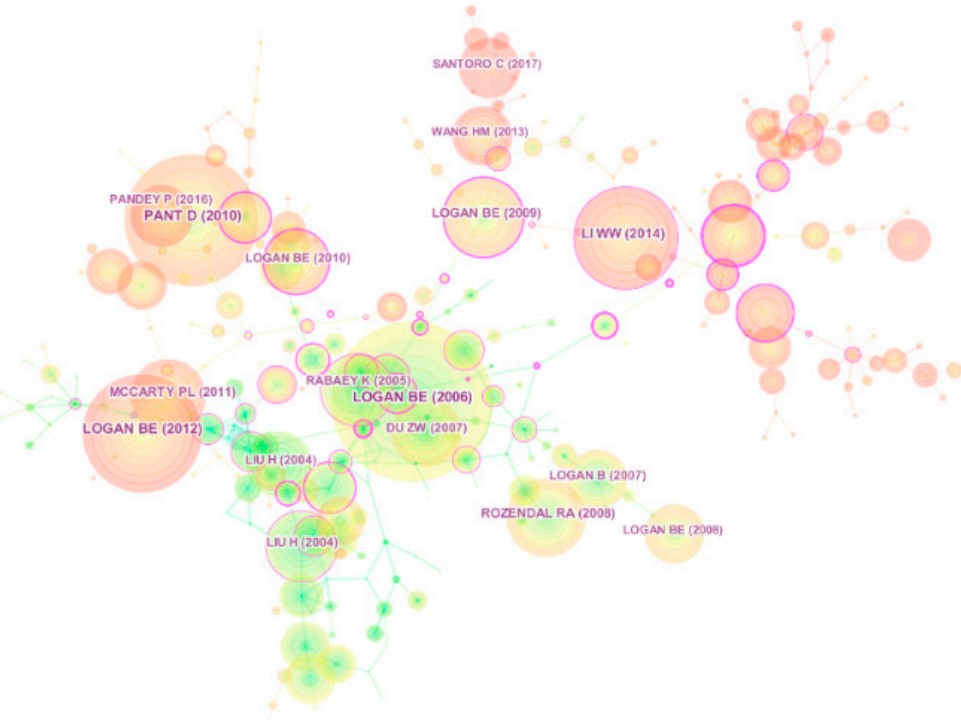

**Figure 4.** Full shot of document co-citation network.

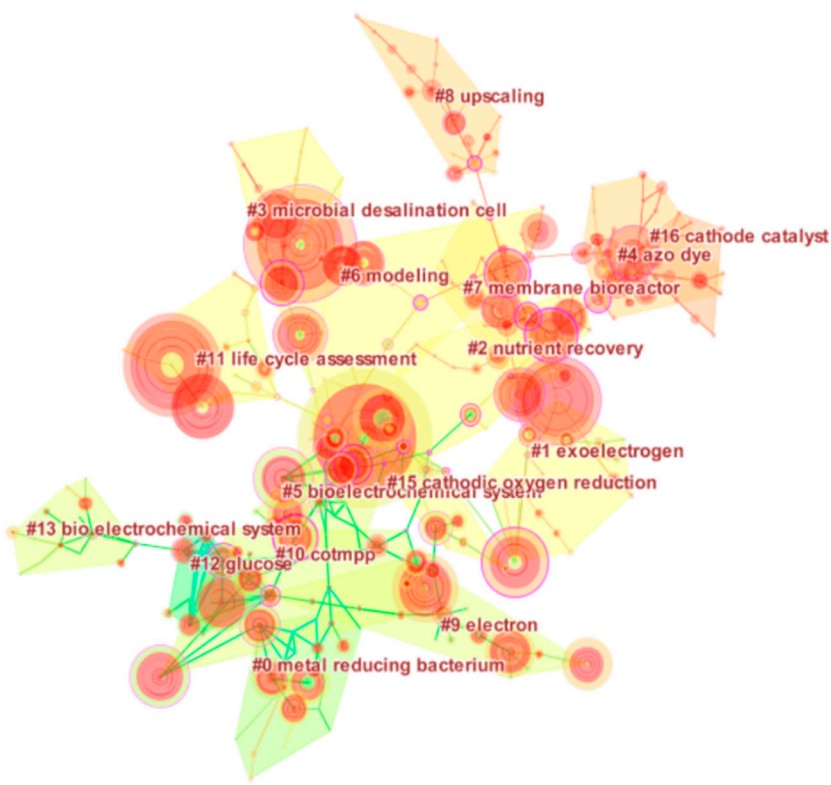

**Figure 5.** Cluster network on MFCs in wastewater treatment.

The top 10 clusters are listed in Table 5. The column of "Size" in Table 5 represents the number of published documents in a cluster, such as the cluster (#0) of "mental reducing bacterium" with 24 members is the biggest one, followed by the cluster (#1, #2, #3 and #4) of "exoelectrogen", "nutrient recovery", "microbial desalination cell" and "azo dye" with 23 members. The sequential of the size indicates the development and the application of MFCs technology are the main research area. The column of "Silhouette" in Table 5 was employed to evaluate the homogeneity of the cluster with the value range of −1.0 to 1.0. All clusters listed in Table 5 obtained a high value of silhouette, indicating better performance in homogeneity.

**Table 5.** Summary of the largest 10 clusters on MFCs in wastewater treatment.

| Cluster ID | Size | Silhouette | Label (TF-IDF) | Mean (Cite Year) |
|:---:|:---:|:---:|:---:|:---:|
| 0 | 24 | 0.936 | Metal reducing bacterium | 2005 |
| 1 | 23 | 0.941 | Exoelectrogen | 2011 |
| 2 | 23 | 0.941 | Nutrient recovery | 2011 |
| 3 | 23 | 0.918 | Microbial desalination cell | 2012 |
| 4 | 23 | 0.937 | Azo dye | 2014 |
| 5 | 22 | 0.866 | Bioelectrochemical system | 2006 |
| 6 | 19 | 0.902 | Modeling | 2011 |
| 7 | 19 | 0.843 | Membrane bioreactor | 2014 |
| 8 | 19 | 0.992 | Upscaling | 2015 |
| 9 | 18 | 0.848 | Electron | 2006 |

The "term frequency-inverse document frequency (TF-IDF)" method was usually employed to evaluate the quality of included documents. TF-IDF was conducted and labeled the cluster in Figure 5 and Table 5.

The index mean (Cite Year) was employed to determine whether the cluster is new or old by analysis of the average year of the published documents. In our study, Cluster

4, Cluster 7 and Cluster 8 were newly formed in 2014 to 2015, indicating "Azo dye", "Membrane bioreactor" and "upscaling" as the hot issues on MFCs in wastewater treatment.

### 4.2. Most Cited References on MFCs in Wastewater Treatment

Table 6 summarizes the top 10 most cited documents on MFCs in wastewater treatment. In Table 6, we found "Microbial Fuel Cells: Methodology and Technology" published by Logan, et al. on *Environmental Science & Technology* as the most cited article. Among the above 10 mentioned cited references, 3 are from *Environmental Science & Technology*, 2 are from *Trends in Biotechnology*, and *Bioresource Technology, Science, Energy & Environmental Science, Nature Reviews Microbiology* and *Biotechnology Advances* each have one.

**Table 6.** The top 10 references on MFCs in wastewater treatment.

| Frequency | Centrality | Title | Authors | Source | Year |
|---|---|---|---|---|---|
| 283 | 0.04 | Microbial Fuel Cells: Methodology and Technology [27] | Logan, B. E. et al. | *Environmental Science & Technology* | 2006 |
| 236 | 0.16 | A review of the substrates used in microbial fuel cells (MFCs) for sustainable energy production [9] | Pant, D. et al. | *Bioresource Technology* | 2010 |
| 211 | 0.04 | Conversion of Wastes into Bioelectricity and Chemicals by Using Microbial Electrochemical Technologies [28] | Logan, B. E. and Rabaey, K. | *Science* | 2012 |
| 180 | 0.12 | Towards sustainable wastewater treatment by using microbial fuel cells-centered technologies [29] | Li, W. W. et al. | *Energy & Environmental Science* | 2014 |
| 143 | 0.01 | Towards practical implementation of bioelectrochemical wastewater treatment [30] | Rozendal, R. A. et al. | *Trends in Biotechnology* | 2008 |
| 141 | 0.32 | Exoelectrogenic bacteria that power microbial fuel cells [31] | Logan, B. E. | *Nature Reviews Microbiology* | 2009 |
| 135 | 0.05 | Domestic wastewater treatment as a Net energy producer—Can this be achieved? [32] | McCarty, P. L. et al. | *Environmental Science & Technology* | 2011 |
| 128 | 0.16 | Microbial fuel cells: novel biotechnology for energy generation [33] | Rabaey, K., and Verstraete, W. | *Trends in Biotechnology* | 2005 |
| 127 | 0.13 | Electricity Generation Using an Air-Cathode Single Chamber Microbial Fuel Cell in the Presence and Absence of a Proton Exchange Membrane [34] | Liu, H. and Logan, B. E. | *Environmental Science & Technology* | 2004 |
| 127 | 0.02 | A state-of-the-art review on microbial fuel cells: A promising technology for wastewater treatment and bioenergy [35] | Du, Z. W. et al. | *Biotechnology Advances* | 2007 |

### 4.3. Top-Tier Journals on MFCs in Wastewater Treatment

The main journals cluster and the top-tier journals on MFCs in wastewater treatment are presented in Figure 6 and Table 7, respectively. From Figure 6, the top-tier journals on MFCs in wastewater treatment are *Environmental Science & Technology, Bioresource Technology, Water Research, Journal of Power Sources, Applied Microbiology and Biotechnology, International Journal of Hydrogen Energy, Water Science and Technology, Applied and Environmental Microbiology* and so on.

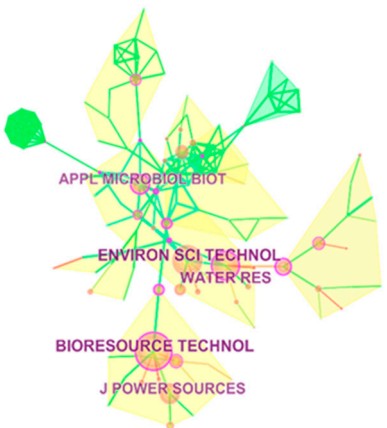

**Figure 6.** Main journals cluster on MFCs in wastewater treatment.

**Table 7.** The top 10 journals according to frequency on MFCs in wastewater treatment.

| Frequency | Centrality | Journals | Half-Life |
|---|---|---|---|
| 2010 | 0.05 | *Environmental Science & Technology* | 13 |
| 1990 | 0.37 | *Bioresource Technology* | 11 |
| 1642 | 0.38 | *Water Research* | 13 |
| 1389 | 0.05 | *Journal of Power Sources* | 10 |
| 1173 | 0.42 | *Applied Microbiology and Biotechnology* | 13 |
| 1072 | 0.02 | *International Journal of Hydrogen Energy* | 12 |
| 1056 | 0.01 | *Biosensors & Bioelectronics* | 13 |
| 970 | 0.27 | *Chemical Engineering Journal* | 10 |
| 879 | 0.00 | *Water Science and Technology* | 11 |
| 848 | 0.43 | *Applied and Environmental Microbiology* | 12 |
| 2010 | 0.05 | *Environmental Science & Technology* | 13 |

## 5. Burst Detection on MFCs in Wastewater Treatment

A burst of an event is a surge of the frequency of the event [36]. Burst detection is a precise analytic tool at the level of references, keywords, authors, and institutions.

### 5.1. Emerging Trends and New Development on MFCs in Wastewater Treatment

The purpose of citation burst detection is to decide whether the article received special attentions from the academic field during the period of study.

In addition, a cluster which was identified as bursts of citations would be accepted as an emerging trend in a certain research field [37]. In our study, we conducted the citation burst detection to estimate the emerging trends and new development on MFCs in wastewater treatment; the top 10 articles with citation bursts are shown in Table 8.

**Table 8.** The top 10 references with strongest citation bursts.

| References | Year | Strength | Begin | End | 1995–2020 |
|---|---|---|---|---|---|
| Bond and Lovley [38] | 2003 | 30.19 | 1995 | 2011 | |
| Chaudhuri and Lovley [39] | 2003 | 23.76 | 1995 | 2011 | |
| Rabaey et al. [40] | 2003 | 23.33 | 1995 | 2011 | |
| Gil et al. [41] | 2003 | 21.62 | 1995 | 2011 | |
| Bond et al. [42] | 2002 | 16.02 | 1995 | 2010 | |
| Kim et al. [43] | 2002 | 16.02 | 1995 | 2010 | |
| Park and Zeikus [44] | 2003 | 15.66 | 1995 | 2011 | |
| Kim et al. [45] | 2004 | 7.77 | 1995 | 2009 | |
| Kim et al. [46] | 1999 | 4.52 | 1995 | 2007 | |
| Park and Zeikus [47] | 2000 | 3.88 | 1995 | 2007 | |

The column of "strength" in Table 8 displayed the time spans character of the networks between each reference. The timeline in Table 8 represented the study period (1995–2020), while the red timeline represented the period of the burst citation references, indicating the duration of hotspot of an attractive thematic.

The top ranked reference by citation bursts is Bond and Lovley [38] in Cluster #12, with bursts of 30.19. The second one is Chaudhuri and Lovley [39] in Cluster #12, with bursts of 23.76. The third is Rabaey et al. [40] in Cluster #12, with burst of 23.33. The fourth is Gil et al. [41] in Cluster #17, with burst of 21.62. The fifth is Bond et al. [42] in Cluster #0, with the burst of 16.02. The sixth is Kim et al. [43] in Cluster #0, with bursts of 16.02. The seventh is Park and Zeikus [44] in Cluster #12, with bursts of 15.66. The eighth is Kim et al. [45] in Cluster #0, with bursts of 7.77. The ninth is Kim et al. [46] in Cluster #17, with bursts of 4.52. The tenth is Park and Zeikus [47] in Cluster #12, with bursts of 3.88. In our study, there are five documents in the top 10 references with strongest citation bursts in the cluster #12. The "glucose", as a main cathode carbon source of MFCs, has become a new hotspot on microbial fuel cells in wastewater treatment.

Burst detection on keywords was usually performed to identify the fast-developing topics in a certain study area [37]. We conducted the burst detection on keywords and summarized the top 10 keyword with bursts in Table 9.

**Table 9.** The top 10 keywords with bursts.

| Keywords | Strength | Begin | End | 1995–2020 |
|---|---|---|---|---|
| Glucose | 16.09 | 1995 | 2010 | |
| Energy | 3.39 | 1995 | 2007 | |
| Biofuel cell | 11.58 | 2006 | 2012 | |
| Hydrogen | 10.93 | 2006 | 2013 | |
| Membrane | 4.71 | 2006 | 2010 | |
| Mediator le | 13.86 | 2007 | 2015 | |
| Continuous electricity generation | 5.14 | 2007 | 2011 | |
| Microbial fuel cell | 19.69 | 2008 | 2014 | |
| Acetate | 4.27 | 2008 | 2012 | |
| Challenge | 3.69 | 2008 | 2010 | |

There is an obvious difference in burst time among the selected keywords listed in Table 9. Beginning in the 1990s, "glucose" and "energy" are the hot research topics on MFCs in wastewater treatment, while there are other keywords bursts after 2010 in this study. Keywords burst may be influenced by several aspects, such as policy, protocol, social issue, environmental pollution incidents and so on. Taking the keyword "energy" for example, the third oil crisis (1990) broke out due to the Gulf War and the price of crude oil nearly doubled. Therefore, it has aroused widespread concern all over the world. This keyword burst has occurred since 1995 and lasted until 2007. The keywords of "MFCs" and "Mediator le" developed as the new topics from 2007 to 2014. Microbial fuel cell is a representative of new biocatalytic green treatment technology with the competence of solving both the energy and water crisis, which is considered to be a promising sustainable technology to meet increasing energy needs [9,48]. That is why the "MFCs" have become a promising technology in wastewater treatment area and has attracted extensive interest from researchers.

*5.2. Burst Detection of Institutions on MFCs in Wastewater Treatment*

Through burst detection on institutions, we can identify the institutions that published a certain quantity of documents in the relevant research field during a certain period of time. The top 10 active institutions were identified by burst detection and are summarized in Table 10.

**Table 10.** The top 10 institutions with bursts.

| Institutions | Strength | Begin | End | 1995–2020 |
|---|---|---|---|---|
| Pennsylvania State University | 10.26 | 1995 | 2014 | |
| The University of Queensland | 7.43 | 2006 | 2011 | |
| National University of Singapore | 4.92 | 2008 | 2012 | |
| Indian Institute of Chemical Technology | 12.09 | 2008 | 2011 | |
| Guangdong Institute of Eco-environment and Soil Sciences | 5.63 | 2009 | 2011 | |
| Peking University | 6.83 | 2009 | 2012 | |
| South China University of Technology | 5.52 | 2010 | 2015 | |
| University of Connecticut | 5.50 | 2010 | 2014 | |
| University of Wisconsin | 8.19 | 2011 | 2014 | |
| University of Science and Technology of China | 5.30 | 2011 | 2014 | |

In this study, the institution of Pennsylvania State University (U.S.A.) first burst in 1995 and lasted 20 years until 2014. This indicates that the authors from Pennsylvania State University (U.S.A.) spent much time and effort on MFCs in wastewater treatment from 1995 to 2014. Of these top 10 institutions with burst strength, four are from China, three are from U.S.A., and Australia, Singapore and India each have one.

*5.3. Burst detection of institutions on MFCs in wastewater treatment*

The burst detection on authors of MFCs usage in wastewater treatment was performed to identify the researchers who published the highest number of articles in a certain period time. The top 10 authors are summarized and listed in Table 11.

**Table 11.** The top 10 authors with bursts.

| Authors (Country) | Strength | Begin | End | 1995–2020 |
|---|---|---|---|---|
| Logan BE (U.S.A.) | 9.07 | 1995 | 2008 | |
| Kim JR (South Korea) | 4.11 | 2005 | 2010 | |
| Keller J (Australia) | 4.53 | 2006 | 2010 | |
| Rabaey K (Australia) | 6.68 | 2006 | 2011 | |
| Ng HY (Singapore) | 5.22 | 2008 | 2012 | |
| Raghavulu SV (India) | 3.94 | 2008 | 2009 | |
| Lefebvre O (Singapore) | 3.65 | 2008 | 2012 | |
| Mohan SV (India) | 12.45 | 2008 | 2012 | |
| Rozendal RA (Australia) | 3.93 | 2008 | 2011 | |
| Mohanakrishna G (India) | 4.23 | 2008 | 2010 | |

In this study, Logan BE, who is recognized as a leading expert on MFCs in wastewater treatment, was identified the first burst author in 1995. Logan BE and his team are from Pennsylvania State University, U.S.A. Of these top 10 authors shown in Table 11, three authors are from Australia and India, Singapore has two, and U.S.A. and South Korea each have one.

## 6. Discussion

We conducted a scientific metrological review on MFCs in wastewater treatment by retrieving 2233 documents from Web of Sciences.

*Environmental Sciences* was identified as having the most popular subject categories on MFCs in wastewater treatment, followed by *Energy Fuels* and *Biotechnology Applied Microbiology*. The journal of *Bioresource Technology* is the most productive journal, having published the highest number of documents related to MFCs in wastewater treatment. Peoples R China, U.S.A., India, South Korea, and Spain published the highest number of documents in this area. As the largest developing country, China is confronted with an energy shortage and environmental pollution. Experts and scholars from China are committed to study the recycling of wastewater. MFC is an environmental and eco-friendly technology that cap-

tures energy through the oxidation of organic substrates in wastewater and converts them into electric current with the help of microorganisms as catalysts [49]. Chinese Academy of Sciences from China, Harbin Institute of Technology from China, Pennsylvania State University from U.S.A., Virginia Polytechnic Institute and State University from U.S.A., and Indian Institute of Technology from India are the five most productive institutions on MFCs in wastewater treatment. Logan BE from Pennsylvania State University is the world's leading expert on MFCs in wastewater treatment.

"Metal reducing bacterium", "Exoelectrogen", "Nutrient recovery", "Microbial desalination cell", "Azo dye", "Bioelectrochemical system", "Modeling", "Membrane bioreactor", "Upscaling" and "Electron" were evaluated as the major research areas of MFCs in wastewater treatment. MFCs can be used to treat wastewater with heavy metals while generating electricity. In recent years, many studies have focused on metal reducing bacterium [50–52]. Becerril-Varela et al. [52] evaluated the feasibility of using an enrichment of iron-reducing bacteria coupled with acetate oxidation to generate electricity in MFC and identified the microorganisms in the MFC. Studies on MFCs in wastewater treatment have also focused on "Exoelectrogen", "Nutrient recovery", "Microbial desalination cell" and "Electron". Li et al. [53] discovered a novel exoelectrogen that can be used to produce electricity and degrade petroleum hydrocarbon. Li et al. [54] demonstrated that the extracellular electron transfer process between electroactive biofilm and electrode is a key step for the performance of MFCs, which is closely related to the enrichment of exoelectrogens and the electrocatalytic activity of the electrode. Recently, MFCs were studied to recover nutrients from wastewater and generate electricity simultaneously [55–57]. Microbial desalination cell has attracted extensive attention in recent years [58–62]. Microbial desalination cell is a low-cost and sustainable option for simultaneously treating wastewater, desalinating saline water, generating electricity, and recovering nutrients from wastewater [63]. Microbial desalination cell can be used to decrease the organic load/COD in wastewater [64]. There are two important mechanisms of extracellular electron transfer, namely direct electron transfer and mediated electron transfer [65]. The exoelectrogens in MFCs utilize metabolism and extracellular electron transfer pathways to obtain energy and generate electricity from organic matter in wastewater [66]. Moreover, "Microbial Fuel Cells: Methodology and Technology" published by Logan, et al. on *Environmental Science & Technology* was selected as the most cited article. *Environmental Science & Technology, Bioresource Technology* and *Water Research* are the most representative journals of MFCs in wastewater treatment.

Furthermore, "Azo dye", "Membrane bioreactor" and "up scaling" were regarded as new emerging research trends on MFCs in wastewater treatment. Azo dyes are one of the most consumed dyes in the world and are used as colorants in food, textile, and pharmaceutical industries [67,68]. It is very urgent to develop efficient, environmentally friendly, and cost-effective technologies for azo dye removal due to the increasingly strict requirements of environmental regulations [69]. In recent years, many studies on azo dye have largely focused on electrode material, operational parameters and reactor configuration [70–72]. MFC integrated with constructed wetland was developed for the treatment of azo dye wastewater [73–75]. Moreover, an MFC-biofilm electrode reactor coupled system was established to promote the degradation of azo dye [76]. Li et al. [77] developed an MFC–microbial electrolysis cell coupled system to promote the decolorization of azo dye and evaluated the effects of anodic substrate concentration and cathodic pH on reactor performance. Li et al. [78] reported that the addition of anthraquinone extracted from natural plants to MFCs can enhance the detoxification and decolorization of azo dyes by mediating electron transfer. An integrated system combining a membrane bioreactor with a MFC was recently developed to augment wastewater treatment [79–81]. More and more studies have focused on MFCs scale-up [82,83]. A scalable composite-engineered electrode module was created for large-scale application and can be used to enhance the decolorization of azo dye wastewater [84,85]. The large-scale application of MFCs has shown good performance in the field of wastewater treatment [86].

## 7. Conclusions

In this paper, the new emerging trends of MFCs in wastewater treatment were evaluated with a scientometric analysis method. A total of 2233 publications were obtained from Web of Sciences during the period 1995–2020. The publication on MFCs in wastewater treatment first appeared in 1995 and the numbers of publication steadily increased every year. This fully explains the research on MFCs in wastewater treatment getting more and more attention. "Environmental Sciences" was identified as the most popular subject categories on MFCs in wastewater treatment and the journal *Bioresource Technology* published the most quantities of articles in this area. China played a leading role in the study of MFCs in wastewater treatment, and Chinese Academy of Science was the most productive institution. "Metal reducing bacterium" was evaluated as the main research area of MFCs in wastewater treatment. More importantly, document co-citation analysis illustrated that the treatment of azo dye wastewater was regarded as the new emerging research trend on MFCs in wastewater treatment.

**Author Contributions:** Conceptualization, T.C.; methodology, C.Z.; software, J.P. and M.W.; validation, T.C. and L.Q.; formal analysis, F.W. and Q.Z.; data curation, C.D.; writing—original draft preparation, Y.Y.; writing—review and editing, J.P. and T.C.; visualization, J.P.; project administration, Y.Y. and H.C.; funding acquisition, T.C. All authors have read and agreed to the published version of the manuscript.

**Funding:** Please add: This research was funded by the Natural Science Foundation of the Jiangsu Higher Education Institutions of China [19KJB610027], by the National Natural Science Foundation of China (NSFC, Grant No. 51608467), by the Key Research and Development Project Special Fund (Social Development) of Jiangsu Province (Grant No. BE2019696), by 'Qing Lan Project' of Colleges and Universities in Jiangsu Province.

**Institutional Review Board Statement:** Not applicable.

**Informed Consent Statement:** Not applicable.

**Data Availability Statement:** Not applicable.

**Acknowledgments:** All authors thank the editor and anonymous reviewers for their constructive comments and suggestions to improve the quality of this paper.

**Conflicts of Interest:** The authors declare no conflict of interest. The funders had no role in the design of the study; in the collection, analyses, or interpretation of data; in the writing of the manuscript; or in the decision to publish the results.

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
