# Peer review of "Mapping Research on Microbial Fuel Cells in Wastewater Treatment: A Co-Citation Analysis"

_processes, doi:10.3390/pr10010179_

Round 1

Reviewer 1 Report

your ideas are good, please, work little bit more to improve English.

The paper is well written in easy to understand terminology.

I think it is an evaluation that tell which journal is addressing which type of articles more. It may help researcher to know the research trends in a particular topic “MFCs’

It is a new idea to see new trends in a research field.

The conclusions are consistent with the evidence and arguments presented

Author Response

Thank you very much for your valuable assessment and comments. We really appreciate your approval of our manuscript. English has been improved.

Reviewer 2 Report

In opinion of this reviewer, this manuscipt is still in a preliminary stage. Althought the information presented is interesting, authors should presented it in a more interesting way. I suggest to the authors organise and discuss the information  deeply.

Author Response

Thank you very much for your valuable assessment and comments. We really appreciate your approval of our manuscript. We have organized and discussed the information in depth as follows:

“We conducted a scientific metrological review on MFCs in wastewater treatment by retrieving 2233 documents from Web of Sciences.

“Environmental Sciences” was identified as the most popular subject categories on MFCs in wastewater treatment and followed by “Energy Fuels” and “Biotechnology Applied Microbiology”. The journal of “Bioresource Technology” in the most productive journal have published the highest number of documents related to MFCs in wastewater treatment. Peoples R China, USA, India, South Korea, and Spain published the highest number of documents in this area. As the largest developing country, China is confronted with energy shortage and environmental pollution. Experts and scholars from China are committed to study the recycling of wastewater. MFC is an environmental and eco-friendly technology that captures energy through the oxidation of organic substrates in wastewater and converts them into electric current with the help of microorganisms as catalysts [49]. Chinese Academy of Sciences from China, Harbin Institute of Technology from China, Pennsylvania State University from USA, Virginia Polytechnic Institute and State University from USA, and Indian Institute of Technology from India are the five most productive institutions on MFCs in wastewater treatment. Logan BE from Pennsylvania State University is the world's leading expert on MFCs in wastewater treatment.

Besides, “Metal reducing bacterium”, “Exoelectrogen”, “Nutrient recovery”, “Microbial desalination cell”, “Azo dye”, “Bioelectrochemical system”, “Modeling”, “Membrane bioreactor”, “Upscaling” and “Electron” were evaluated as the major research areas of MFCs in wastewater treatment. MFCs can be used to treat wastewater with heavy metals while generating electricity. In recent years, many studies have focused on metal reducing bacterium [50-52]. Becerril-Varela et al. (2021) evaluated the feasibility of using an enrichment of iron reducing bacteria coupled with acetate oxidation to generate electricity in MFC and identified the microorganisms in the MFC. Besides, studies on MFCs in wastewater treatment have also focused on “Exoelectrogen”, “Nutrient recovery”, and “Microbial desalination cell”. Li et al. (2021) demonstrated that the extracellular electron transfer process between electroactive biofilm and electrode is a key step for the performance of MFCs, which is closely related to the enrichment of exoelectrogens and the electrocatalytic activity of the electrode [53]. Recently, MFCs have been studied to recover nutrients from wastewater and generate electricity simultaneously [54-56]. Microbial desalination cell is a low-cost and sustainable option for simultaneously treating wastewater, desalinating saline water, generating electricity, and recovering nutrients from wastewater [57]. Moreover, “Microbial Fuel Cells: Methodology and Technology” published by Logan, et al. on Environmental Science & Technology was selected as the most cited article. “Environmental Science & Technology”, “Bioresource Technology” and “Water Research” are the most representative journals of MFCs in wastewater treatment.

Furthermore, “Azo dye”, “Membrane bioreactor” and “up scaling” were regarded as new emerging research trend on MFCs in wastewater treatment. In recent years, many studies on azo dye have largely focused on electrode material, operational parameters and reactor configuration [58-60]. MFC integrated with constructed wetland has been developed for the treatment of azo dye wastewater [61-62]. An integrated system combining a membrane bioreactor with a MFC has recently been developed to augment wastewater treatment [63-65]. More and more studies have focused on MFCs scale-up [66-67]. The large-scale application of MFCs have shown good performance in the field of wastewater treatment [68].” (page 11-12, line 276-318 in the revised manuscript)

Reviewer 3 Report

The mapping research on microbial fuel cells in wastewater treatment is hown in this paper. In this manuscript a co-citation analysis was analysed.

The suggestions or questions are listed below:
1. The poor discussion of the results. It would be good to transfer the information (line 269-289) from Conclusion chapter to Disscusion chapter (which should be added).

2. It is necessary to rewrite Conclusion chapter.
- line 290-292 - the sentence is unnecessary. It is up to the readers to judge whether this analysis will be an important source of information for them or not. If so, it will be reflected in the citations.
- line 292-295 - it is not an conclusion of review. It sentence can be used in Abstract chapter or in end of Introduction chapter.
The Conclusion chapter should contain conclusions from the information contained in line 269-289. For example, on the basis of citations, it should be concluded what are the current trends in the development of MFC technology.

3.References: There are 48 references in the reference list. Definitely too little references for a review article. Such a number of references would be appropriate in a research work, where it is the recognition of the subject for conducting own research. 
It is necessary to expand the number of analyzed references and include them in the conducted analysis.

4. It should be good to  increase the resolution of figures.

Author Response

Thank you very much for your valuable assessment and comments. We really appreciate your approval of our manuscript. The revisions according to your suggestions and comments are shown as follows:

Q1. The poor discussion of the results. It would be good to transfer the information (line 269-289) from Conclusion chapter to Discussion chapter (which should be added).

Answer: Thanks for your valuable suggestions. We have added the Discussion chapter. We have transfered the information (line 269-289) from Conclusion chapter to Disscusion chapter. We have also added the relevant discussion (page 11-12, line 276-318 in the revised manuscript) as follows:

“We conducted a scientific metrological review on MFCs in wastewater treatment by retrieving 2233 documents from Web of Sciences.

“Environmental Sciences” was identified as the most popular subject categories on MFCs in wastewater treatment and followed by “Energy Fuels” and “Biotechnology Applied Microbiology”. The journal of “Bioresource Technology” in the most productive journal have published the highest number of documents related to MFCs in wastewater treatment. Peoples R China, USA, India, South Korea, and Spain published the highest number of documents in this area. As the largest developing country, China is confronted with energy shortage and environmental pollution. Experts and scholars from China are committed to study the recycling of wastewater. MFC is an environmental and eco-friendly technology that captures energy through the oxidation of organic substrates in wastewater and converts them into electric current with the help of microorganisms as catalysts [49]. Chinese Academy of Sciences from China, Harbin Institute of Technology from China, Pennsylvania State University from USA, Virginia Polytechnic Institute and State University from USA, and Indian Institute of Technology from India are the five most productive institutions on MFCs in wastewater treatment. Logan BE from Pennsylvania State University is the world's leading expert on MFCs in wastewater treatment.

Besides, “Metal reducing bacterium”, “Exoelectrogen”, “Nutrient recovery”, “Microbial desalination cell”, “Azo dye”, “Bioelectrochemical system”, “Modeling”, “Membrane bioreactor”, “Upscaling” and “Electron” were evaluated as the major research areas of MFCs in wastewater treatment. MFCs can be used to treat wastewater with heavy metals while generating electricity. In recent years, many studies have focused on metal reducing bacterium [50-52]. Becerril-Varela et al. (2021) evaluated the feasibility of using an enrichment of iron reducing bacteria coupled with acetate oxidation to generate electricity in MFC and identified the microorganisms in the MFC. Besides, studies on MFCs in wastewater treatment have also focused on “Exoelectrogen”, “Nutrient recovery”, and “Microbial desalination cell”. Li et al. (2021) demonstrated that the extracellular electron transfer process between electroactive biofilm and electrode is a key step for the performance of MFCs, which is closely related to the enrichment of exoelectrogens and the electrocatalytic activity of the electrode [53]. Recently, MFCs have been studied to recover nutrients from wastewater and generate electricity simultaneously [54-56]. Microbial desalination cell is a low-cost and sustainable option for simultaneously treating wastewater, desalinating saline water, generating electricity, and recovering nutrients from wastewater [57]. Moreover, “Microbial Fuel Cells: Methodology and Technology” published by Logan, et al. on Environmental Science & Technology was selected as the most cited article. “Environmental Science & Technology”, “Bioresource Technology” and “Water Research” are the most representative journals of MFCs in wastewater treatment.

Furthermore, “Azo dye”, “Membrane bioreactor” and “up scaling” were regarded as new emerging research trend on MFCs in wastewater treatment. In recent years, many studies on azo dye have largely focused on electrode material, operational parameters and reactor configuration [58-60]. MFC integrated with constructed wetland has been developed for the treatment of azo dye wastewater [61-62]. An integrated system combining a membrane bioreactor with a MFC has recently been developed to augment wastewater treatment [63-65]. More and more studies have focused on MFCs scale-up [66-67]. The large-scale application of MFCs have shown good performance in the field of wastewater treatment [68].”

  1. It is necessary to rewrite Conclusion chapter.

(1)- line 290-292 - the sentence is unnecessary. It is up to the readers to judge whether this analysis will be an important source of information for them or not. If so, it will be reflected in the citations.

Answer: Thanks for your valuable suggestions. The sentence has been deleted.

(2)- line 292-295 - it is not an conclusion of review. It sentence can be used in Abstract chapter or in end of Introduction chapter.

Answer: Thanks for your valuable suggestions. The sentences have been used in end of Introduction chapter.

“With the aim to explore the key literatures and major research area on MFCs in wastewater treatment, we summarized the published information in this field from the Web of Science Core Collection in the past 25 years (1995-2020) and conducted co-citation analysis to identify the core literatures and investigate the major research area on MFCs in wastewater treatment by CiteSpace. This study estimates the most important and remarkable research trends and hot spots of MFCs usage in wastewater treatment. In addition, this study provides an overview of the development of MFCs usage in wastewater treatment and also offers valuable suggestions to energy and environmental management.” (page 2, line 77-80 in the revised manuscript)

(3) The Conclusion chapter should contain conclusions from the information contained in line 269-289. For example, on the basis of citations, it should be concluded what are the current trends in the development of MFC technology.

Answer: Thanks for your valuable suggestions. We have rewritten the Conclusion chapter according to your suggestions and comments (page 12, line 320-331 in the revised manuscript) as follows:

“In this paper, the new emerging trends of MFCs in wastewater treatment was evaluated with a scientometric analysis method. A total of 2233 publications were obtained from Web of Sciences during the period 1995-2020. The publication on MFCs in wastewater treatment first appeared in 1995 and the numbers of publication steadily increased every year. This fully explains the researches on MFCs in wastewater treatment are getting more and more attention. “Environmental Sciences” was identified as the most popular subject categories on MFCs in wastewater treatment and the journal “Bioresource Technology” published the most quantities of articles in this area. China played a leading role in the study of MFCs in wastewater treatment and Chinese Academy of Science was the most productive institution. “Metal reducing bacterium” was evaluated as the main research areas of MFCs in wastewater treatment. More importantly, document co-citation analysis illustrated that the treatment of Azo dye wastewater was regarded as the new emerging research trend on MFCs in wastewater treatment.”

3.References: There are 48 references in the reference list. Definitely too little references for a review article. Such a number of references would be appropriate in a research work, where it is the recognition of the subject for conducting own research.

It is necessary to expand the number of analyzed references and include them in the conducted analysis.

Answer: Thanks for your valuable suggestions. We have added some relevant discussion and the related references have been added in the reference lists:

“49. Mohyudin S, Farooq R, Jubeen F, et al. Microbial fuel cells a state-of-the-art technology for wastewater treatment and bioelectricity generation. Environ. Res 2022; 204:112387.

  1. Groudev S, Spasova I, Groudeva V, et al. Passive treatment of metal-polluted waters in combination with electricity generation by microbial fuel cells. Comptes Rendus L’Academie Bulg 2020; 73:73–81.
  2. Wu Y, Wang L, Jin M, et al. Simultaneous copper removal and electricity production and microbial community in mi-crobial fuel cells with different cathode catalysts. Bioresour. Technol 2020; 305:123166.
  3. Becerril-Varela K, Serment-Guerrero J H, Manzanares-Leal G L, Generation of electrical energy in a microbial fuel cell coupling acetate oxidation to Fe3+ reduction and isolation of the involved bacteria. World J. Microbiol. Biotechnol 2021; 37:1-15.
  4. Li Y, Liu J, Chen X, et al. Tailoring Surface Properties of Electrodes for Synchronous Enhanced Extracellular Electron Transfer and Enriched Exoelectrogens in Microbial Fuel Cells. ACS Appl. Mater. Interfaces 2021; 13:58508–58521.
  5. Ye Y, Ngo H H, Guo W, et al. Effect of organic loading rate on the recovery of nutrients and energy in a dual-chamber microbial fuel cell. Bioresour. Technol 2019; 281:367-373.
  6. Ye Y, Ngo H H, Guo W, et al. Microbial fuel cell for nutrient recovery and electricity generation from municipal wastewater under different ammonium concentrations. Bioresour. Technol 2019; 292:121992.
  7. Paucar N E, Sato C. Microbial fuel cell for e. nergy production, nutrient removal and recovery from wastewater: A review. Processes 2021; 9:1318.
  8. Gujjala L K S, Dutta D, Sharma P, et al. A state-of-the-art review on microbial desalination cells. Chemosphere 2022; 288:132386.
  9. Long X, Wang H, Wang C, et al. Enhancement of azo dye degradation and power generation in a photoelectrocatalytic microbial fuel cell by simple cathodic reduction on titania nanotube arrays electrode. J. Power Sources 2019; 415:145-153.
  10. Ilamathi R, Sheela A M, Performance analysis of microbial fuel cell operational parameters on reactive azo dye decol-orization. Desalin. Water Treat. 2020; 190:312-321.
  11. Oon Y S, Ong S A, Ho L N, et al. Innovative baffled microbial fuel cells for azo dye degradation: Interactive mechanisms of electron transport and degradation pathway. J. Clean. Prod. 2021; 295:126366.
  12. Oon Y L, Ong S A, Ho L N, et al. Up-flow constructed wetland-microbial fuel cell for azo dye, saline, nitrate remediation and bioelectricity generation: From waste to energy approach. Bioresour. Technol. 2018; 266:97-108.
  13. Oon Y L, Ong S A, Ho L N, et al. Constructed wetland–microbial fuel cell for azo dyes degradation and energy recovery: Influence of molecular structure, kinetics, mechanisms and degradation pathways. Sci. Total Environ 2020; 720:137370.
  14. Ibrahim R S B, Zainon Noor Z, Baharuddin N H, et al. Microbial Fuel Cell Membrane Bioreactor in Wastewater Treatment, Electricity Generation and Fouling Mitigation. Chem. Eng. Technol 2020; 43:1908-1921.
  15. Li T, Cai Y, Yang X, et al. Microbial Fuel Cell-Membrane Bioreactor Integrated System for Wastewater Treatment and Bioelectricity Production: Overview. J. Environ. Eng 2020; 146:04019092.
  16. Zhao S, Yun H, Khan A, et al. Two-stage microbial fuel cell (MFC) and membrane bioreactor (MBR) system for enhancing wastewater treatment and resource recovery based on MFC as a biosensor. Environ. Res. 2022; 204:112089.
  17. Goto Y, Yoshida N. Scaling up microbial fuel cells for treating swine wastewater. Water 2019; 11:1803.
  18. Jadhav D A, Mungray A K, Arkatkar A, et al. Recent advancement in scaling-up applications of microbial fuel cells: From reality to practicability. Sustain. Energy Technol. Assessments 2021; 45:101226.
  19. Tan W H, Chong S, Fang H W, et al. Microbial fuel cell technology-a critical review on scale-up issues. Processes 2021; 9:985.” (page 13-15, line 448-487 in the revised manuscript).
  20. It should be good to increase the resolution of figures.

Answer: Thanks for your valuable suggestions. We have increased the resolution of figures.

Round 2

Reviewer 3 Report

The Authors revised the manuscript according to the comments of the reviewer.
However, due to the fact that this is a Review, it would be nice to increase the references by a few more items. This will increase the substantive value of the work.

Author Response

Answer: Thanks for your valuable suggestions. We have added some relevant discussion. The related references have been added in the reference lists:

Li et al. [53] discovered a novel exoelectrogen that can be used to produce electricity and degrade petroleum hydrocarbon. Li et al. [54] demonstrated that the extracellular electron transfer process between electroactive biofilm and electrode is a key step for the performance of MFCs, which is closely related to the enrichment of exoelectrogens and the electrocatalytic activity of the electrode. Recently, MFCs have been studied to recover nutrients from wastewater and generate electricity simultaneously [55-57]. Microbial desalination cell has attracted extensive attention in recent years [58-62]. Microbial desalination cell is a low-cost and sustainable option for simultaneously treating wastewater, desalinating saline water, generating electricity, and recovering nutrients from wastewater [63]. Microbial desalination cell can be used to decrease the organic load/COD in wastewater [64]. There are two important mechanisms of extracellular electron transfer, namely direct electron transfer and mediated electron transfer [65]. The exoelectrogens in MFCs utilize metabolism and extracellular electron transfer pathways to obtain energy and generate electricity from organic matter in wastewater [66]. Moreover, “Microbial Fuel Cells: Methodology and Technology” published by Logan, et al. on Environmental Science & Technology was selected as the most cited article. “Environmental Science & Technology”, “Bioresource Technology” and “Water Research” are the most representative journals of MFCs in wastewater treatment.

Furthermore, “Azo dye”, “Membrane bioreactor” and “up scaling” were regarded as new emerging research trend on MFCs in wastewater treatment. Azo dyes are one of the most consumed dyes in the world and are used as colorants in food, textile, and pharmaceutical industries [67-68]. It is very urgent to develop efficient, environmental-friendly, and cost-effective technologies for azo dye removal due to the increasingly strict requirements of environmental regulations [69]. In recent years, many studies on azo dye have largely focused on electrode material, operational parameters and reactor configuration [70-72]. MFC integrated with constructed wetland has been developed for the treatment of azo dye wastewater [73-75]. Moreover, a MFC-biofilm electrode reactor coupled system was established to promote the degradation of the azo dye [76]. Li et al. [77] developed a MFC–microbial electrolysis cell coupled system to promote the decolourization of azo dye and evaluated the effects of anodic substrate concentration and cathodic pH on reactor performance. Li et al. [78] reported that the addition of anthraquinone extracted from natural plants to MFCs can enhance the detoxification and decolorization of azo dyes by mediating electron transfer. An integrated system combining a membrane bioreactor with a MFC has recently been developed to augment wastewater treatment [79-81]. More and more studies have focused on MFCs scale-up [82-83]. A scalable composite-engineered electrode module was created for large-scale application and can be used to enhance the decolourization of azo dye wastewater [84-85]. The large-scale application of MFCs has shown good performance in the field of wastewater treatment [86]. (page 12, line 300-335 in the revised manuscript)

References

  1. Li X, Zheng R, Zhang X, et al. A novel exoelectrogen from microbial fuel cell: Bioremediation of marine petroleum hydrocarbon pollutants. J. Environ. Manage 2019; 235:70-76.
  2. Ewusi-Mensah D, Huang J, Chaparro L K, et al. Algae-assisted microbial desalination cell: Analysis of cathode performance and desalination efficiency assessment. Processes 2021; 9:2011.
  3. Goren A Y, Okten H E. Energy production from treatment of industrial wastewater and boron removal in aqueous solutions using microbial desalination cell. Chemosphere 2021; 285:131370.
  4. Goren A Y, Okten H E. Simultaneous energy production, boron and COD removal using a novel microbial desalination cell. Desalination 2021; 518:115267.
  5. Rahman S, Al-Mamun A, Jafary T, et al. Effect of internal and external resistances on desalination in microbial desalination cell. Water Sci. Technol 2021; 83:2389-2403.
  6. Ramírez-Moreno M, Esteve-Núñez A, Ortiz J M, 2021. Desalination of brackish water using a microbial desalination cell: Analysis of the electrochemical behaviour. Electrochim. Acta 2021; 388:138570.
  7. Zahid M, Savla N, Pandit S, et al. Microbial desalination cell: Desalination through conserving energy. Desalination 2022; 521:115381.
  8. Prathiba S, Kumar P S, Vo D V N. Recent advancements in microbial fuel cells: A review on its electron transfer mechanisms, microbial community, types of substrates and design for bio-electrochemical treatment. Chemosphere 2022; 286:131856.
  9. Li Y, Liu J, Chen X, et al. Tailoring spatial structure of electroactive biofilm for enhanced activity and direct electron transfer on iron phthalocyanine modified anode in microbial fuel cells. Biosens. Bioelectron 2021; 191:113410.
  10. Cui M H, Sangeetha T, Gao L, et al. Efficient azo dye wastewater treatment in a hybrid anaerobic reactor with a built-in integrated bioelectrochemical system and an aerobic biofilm reactor: Evaluation of the combined forms and reflux ratio. Bioresour. Technol 2019; 292:122001.
  11. Cui D, Cui M H, Liang B, et al. Mutual effect between electrochemically active bacteria (EAB) and azo dye in bio-electrochemical system (BES). Chemosphere 2020; 239:124787.
  12. Cui M H, Liu W Z, Tang Z E, et al. Recent advancements in azo dye decolorization in bio-electrochemical systems (BESs): Insights into decolorization mechanism and practical application. Water Res 2021; 203:117512.
  13. Fang Z, Song H, Yu R, et al. A microbial fuel cell-coupled constructed wetland promotes degradation of azo dye decolorization products. Ecol. Eng 2016; 94:455-463.
  14. Cao X, Wang H, Li X, et al. Enhanced degradation of azo dye by a stacked microbial fuel cell-biofilm electrode reactor coupled system. Bioresour. Technol 2017; 227:273-278.
  15. Li Y, Yang H Y, Shen J Y, et al. Enhancement of azo dye decolourization in a MFC-MEC coupled system. Bioresour. Technol 2016; 202:93-100.
  16. Li T, Song H L, Xu H, et al. Biological detoxification and decolorization enhancement of azo dye by introducing natural electron mediators in MFCs. J. Hazard. Mater 2021; 416:125864.
  17. Wang A, Wang H, Cheng H, et al. Electrochemistry-stimulated environmental bioremediation: Development of applicable modular electrode and system scale-up. Environ. Sci. Ecotechnology 2020; 3:100050.
  18. Wang A, Shi K, Ning D, et al. Electrical selection for planktonic sludge microbial community function and assembly. Water Res. 2021; 206:117744.
